# ESP: Exponential Smoothing on Perturbations for Increasing Robustness to Data Corruptions

## Abstract

Despite the great advances in the machine learning field over the past decade, deep learning algorithms are often vulnerable to data corruption in real-world environments. We propose a simple yet efficient data augmentation method named Exponential Smoothing on Perturbations (ESP) that imposes perturbations on training data to enhance a model's robustness to unforeseen data corruptions. With the perturbation on the input side, the target label of a sample is smoothed with an exponentially decaying confidence level with respect to the size of the perturbation. ESP enforces a contour-like decision boundary that smoothly encompasses the region around inter-class samples. We theoretically show that perturbations in input space can encourage a model to find a flat minimum on the parameter space, which makes a model robust to domain shifts. In the extensive evaluation on common corruption benchmarks including MNIST-C, CIFAR-10/100-C, and Tiny-ImageNet-C, our method improves the robustness of a model both as a standalone method and in conjunction with the previous state-of-the-art augmentation-based methods. ESP is a model-agnostic algorithm in the sense that it is neither model-specific nor data-specific.

## 1 Introduction

Over the past decade, deep learning models have rapidly evolved to update state-of-the-art performance on a wide range of machine learning tasks, including computer vision, natural language processing, reinforcement learning, etc. Despite the remarkable advances in learning algorithms, deep models are often prone to data corruptions that hinder the successful training of networks. Albeit the importance of robust training, it is very recent that the robustness of deep models to real-world-driven data corruption has gained attention in the machine learning society. The vulnerability of the deep neural network (DNN) against adversarial perturbations was first raised way back in the early 2010s (Szegedy et al., 2013; Goodfellow et al., 2015), and numerous methods have been proposed to enhance the model's robustness since then (Cui et al., 2021; Salman et al., 2020; Madry et al., 2018). On the other hand, the benchmarks for evaluating DNN's robustness to real-world driven common corruptions such as noise, blur, fog, etc., have been only recently established for the image classification tasks (Hendrycks & Dietterich, 2019; Mu & Gilmer, 2019), and algorithms to improve the model robustness against the common corruptions are at their early stage of development (Hendrycks et al., 2021b;a; Rusak et al., 2020; Wang et al., 2021a).

Recent approaches for improving the robustness to common corruptions in the image classification tasks either utilize image augmentation methods (Hendrycks et al., 2021b;a; Rusak et al., 2020; Calian et al., 2021), propose novel model architectures (Kim et al., 2021; Mao et al., 2021; He et al., 2021), or adopt the adaptation learning settings (Wang et al., 2021a; Rusak et al., 2021). While it is not yet revealed what the most dominant strategy against common corruption is, the group of augmentation-based methods shares the desirable property that it can be easily combined with other promising methods to further enhance the model's robustness. In this aspect, augmentation-based methods can be regarded as model-agnostic algorithms with a wide range of applicability. In addition, empirical evidence demonstrates that exploiting diverse data augmentations can effectively enhance the model's robustness against common corruptions in many real-world scenarios (Hendrycks et al.,

Figure 1: Visualization of the decision boundaries of the classifiers trained with the original dataset (denoted as 'Naive'), the augmented dataset with random noise with fixed $L_2$ distance (denoted as '$L_2$'), and the augmented dataset with our method (denoted as ESP). The spiral-like distributed inputs from two classes are generated for the original dataset. The decision boundaries of the different classifiers are illustrated. **(a)** ESP enforces a contour-like decision boundary which generalizes better than $L_2$. **(b)** ESP is less sensitive to the maximum perturbation size compared to $L_2$.

2021a; Calian et al., 2021). However, there has been weak theoretical understanding on how such data augmentations can enhance the model robustness.

We propose a method named Exponential Smoothing on Perturbations (ESP) that introduces the data perturbation in the form of $L_2$ distance-based stochastic noises on the input space. Also, ESP smoothes the confidence level of the target label for the perturbed input to be decaying with respect to the size of the perturbation. In addition, we theoretically show that input perturbations that have bounded $L_2$ norm can make a model find flatter minima in the parameter space. A model with flat minima has a strong domain generalization capability (Cha et al., 2021a) and robustness to adversarial examples (Stutz et al., 2021).

In the extensive simulations on the common corruption benchmarks, including MNIST-C (Mu & Gilmer, 2019) and CIFAR-10/100-C (Hendrycks & Dietterich, 2019), and Tiny-ImageNet-C, a standalone ESP or a combined model in conjunction with prior data augmentation methods achieves state-of-the-art accuracies with considerable margins. The main contribution of this paper is threefold:

- We provide a new perspective on label smoothing (Szegedy et al., 2016) as a tool to embed the uncertainty of data perturbations in the input space. Furthermore, we show that the optimal decision boundary formed by the label smoothing function of ESP makes the classifier more affected by the topologies of manifolds and less affected by the number of datapoints in manifolds.

- We demonstrate that our method ESP, is at least effective as $L_2$ distance based perturbations both empirically and theoretically. With minimal assumptions and implications on the nature of datasets and models, ESP is shown to improve model robustness in common corruption benchmarks in all experiment cases further than $L_2$ noise.

- We analyze how perturbations in input space can be related to perturbations in parameter space. It has been proved that finding flat minima in the parameter space makes classifier robust against distribution shifts in the test dataset (Cha et al., 2021b). We partially formalize the above idea by considering how the perturbation regions in the input space and parameter space can be related to each other via a linear model.

## 2 RELATED WORK

Recently, various strategies for enhancing model robustness have been suggested. Here, we categorize prior methods into three types: data augmentation-based, model-specific, and adaptation-based approaches.

**Data Augmentation-based Approaches** The most popular approach to increase model robustness is augmenting training data to mimic the corruptions as a form of data transformation. AugMix of Hendrycks et al. (2021b) is an image augmentation method that composes randomly-sampled basic image processing operations to produce a novel image that maintains the semantic information of the original image sample. In the training phase, AugMix utilizes the generated novel image samples located around the original sample in the input space. To be specific, the divergence between the

posterior distributions of the original and augmented samples is minimized for a model to embed the augmented samples close to the original one. When we compare Exponential Smoothing on Perturbations (ESP) with AugMix, our method adopts a simpler form of augmentation with $L_2$-norm-bounded perturbations on the input space and directly trains the augmented sample with the exponentially smoothed soft label.

Another group of approaches leverages parameterized models for data augmentation. DeepAugment of Hendrycks et al. (2021a) distorts the image using a pretrained image-to-image transforming model to generate augmented images. Besides, DeepAugment adopts the perturbations on networks by employing predefined processes of hidden signals such as zeroing, negating, transposing, etc. The main differences between ESP and DeepAugment are the utilization of parameterized augmentation methods and the perturbations on networks. In the view of augmentation, ESP does not require additional deep networks for transforming original images. In our theoretical analysis, we provide an insight that relates the bounded perturbations on input via ESP to the perturbations in the parameter space. On the other hand, DeepAugment introduces model perturbations by processing the hidden signals, which are not restricted to the form of $L_2$-norm bounded perturbation in the parameter space. Also, the robustness of DeepAugment is not theoretically guaranteed. Adversarial Noise Training (ANT) of Rusak et al. (2020) trains an additional noise generator that produces adversarial noise that maximally confuses the classifier. ANT is related to our method where they focus on imposing the noise-based perturbations on the input space. However, ESP explicitly smooths the target label with respect to the size of the perturbation and does not require additional training of a noise generator. ESP is neither model-specific nor data-specific when compared to ANT which relies on model- and data-specific noise generator.

Mixup of Zhang et al. (2018) linearly interpolates between two data points from different classes and trains a classification model on the dataset that includes combined samples. The interpolated image is labeled by the interpolation between two one-hot labels of the original data samples. Mixup differs from our method, where the two samples are interpolated to construct a novel training sample. From the perspective of the label smoothing by ESP, Mixup also smooths the target label of the combined sample by interpolating the original labels. However, Mixup suffers from the manifold intrusion problem due to the conflict between the interpolated manifold and other original manifolds (Hendrycks et al., 2021b). ESP can alleviate the manifold intrusion problem by choosing a proper perturbation size and the degree of smoothing on the input space.

**Model Architecture-based Approaches** Another branch of approaches is developing a model-specific training scheme. Based on the clean image samples, QualNet of Kim et al. (2021) pretrains a classifier with the invertible architecture and inverts it to obtain a decoder that is capable of reconstructing original images from the corresponding feature vectors. The prepared decoder is used as a reconstruction module that takes the features from a new target classifier to be trained in the second stage. Even from the corrupted input samples with low quality, the target classifier is then trained to construct clean-like features that can be decoded into high-quality images. Vision transformer of (Dosovitskiy et al., 2021) has recently gained attention in building a robust vision classifier. Some works have changed the components of vision transformers to gain robustness (Mao et al., 2021; Zhou et al., 2022; Mao et al., 2022), while others have designed self-supervised tasks for vision transformers (He et al., 2021). Despite the fact that the vision transformer-based approaches have been continuously updating their remarkable performance on common corruption benchmarks, they suffer from deficient generalizability. The group of model-specific methods relies on carefully designed model architectures so that they have limitations to be combined with other methods.

**Domain Adaptation-based Approaches** The other approach borrows the concept of domain adaptation to improve model robustness. Test Entropy Minimization (TENT) of Wang et al. (2021a) is a domain adaptation method that tunes the parameters of the batch normalization layers in the test time. The adaptation method indeed enhances the generalization capability to the common corruption that can be considered as input domain shifts. Robust Pseudo Labeling (RPL) of Rusak et al. (2021) assumes the unsupervised domain adaptation setting and exploits a self-learning method for training classifiers. The branch of adaptation-based methods requires additional access to the target data either at the training stage (domain adaptation) or at the test time (test-time adaptation), which makes their usage restricted to specific circumstances.

## 3 ESP: EXPONENTIAL SMOOTHING ON PERTURBATIONS

### 3.1 BACKGROUND AND MOTIVATIONS OF ESP

Herein, we present the background and motivations for the algorithmic details of ESP.

**Specifications of $L_2$-Norm-based Noise** When adding $L_2$ distance-based random noise to an input, the perturbed input often lies outside the valid input domain, e.g., $[0, 1]^{28 \times 28}$ for samples in MNIST dataset. A simple clipping of the perturbed sample into the valid input domain probably results in a smaller effective noise than desired. To cope with the problem, we utilize the noise rescaling and clipping algorithm of (Rauber & Bethge, 2020) that preserves the desired $L_2$-norm of the noise while restricting the perturbed input to the valid domain. For the sake of simplicity, we will denote the rescaled and clipped $L_2$ distance-based random noise simply as $L_2$ noise henceforth.

**Desired Properties of $L_2$ Noise** Too large noise probably intrudes on other classes. When thinking of a perturbed input sample that moves far away from the original data point, the noised sample can intrude on other class manifolds so that the model robustness eventually decreases. In Figure 1b, $L_2$ noise with an excessive amount severely deteriorates the training of classifiers. However, reducing the size of $L_2$ noise raises another issue. The augmented samples should locate effectively far away from the original data point to guarantee sufficient margins of decision boundaries. To this end, we utilize a truncated Gaussian distribution with non-zero mean $\epsilon > 0$ to sample the power of $L_2$ noise.

**Desired Properties of Label Smoothing** Label smoothing is conventionally exploited for model calibration and penultimate layer's equidistant embedding in a static fashion (Müller et al., 2019). Label smoothing assigns $1 - \alpha$ for the true label and $\alpha/(C-1)$ for the other labels, where $\alpha \in (0, 1)$ is a constant hyperparameter and $C$ is the number of classes. On the other hand, we re-purpose the label smoothing technique as a tool for embedding the uncertainty of perturbations in the input space and defining $\alpha$ as a perturbation size-dependent hyperparameter. By giving stronger label smoothing to larger perturbations, the decision boundaries of a classifier are less affected by the number of perturbed data points but more by the distribution of data points. Nonetheless, as shown in Figure 1a, the application of smoothing from the region nearby the original data point can sharply shape the decision boundaries without sufficient margins.

Our method, ESP processes probabilistic samplings of perturbation size and dynamic label smoothing functions that are carefully chosen. In the following sections, we formally describe our data augmentation strategy that guarantees the aforementioned properties of $L_2$ noise and label smoothing.

### 3.2 ALGORITHMIC DETAILS OF ESP

ESP consists of three components: random orientation sampling, random size sampling, and a smoothing function. First, the orientation of the perturbation vector, i.e., the normalized directional vector of the perturbation, is randomly sampled. Since isotropic Gaussian has equal probability over the vector orientations, we have implemented the random orientation sampling by sampling a Gaussian vector and normalizing it. Second, the size of perturbation vector is sampled with a pre-defined probability density function. In our experiments, truncated Gaussian distribution is used. Finally, the hard label of the a datapoint is smoothed with respect to the $L_2$ norm, or the size of the perturbation vector. While any arbitrary nonincreasing function can be used, we use an exponentially decaying function to smooth the original label, after a certain threshold $\epsilon$ of the perturbation size. One reason using exponential function $ae^{-\lambda x}$ is that for every $x_1$ and $x_2$, $ae^{-\lambda(x_1+b)}/ae^{-\lambda x_1} = ae^{-\lambda(x_2+b)}/ae^{-\lambda x_2} = ae^{-\lambda b}$ holds. In other words, the original label is smoothed exponentially as the perturbation size grows, and the extent of smoothing is solely dependent on the relative sizes of perturbations.

There are total four hyperparameters consisting of ESP, three for the truncated Gaussian and one for the smoothing function. To be more specific, the truncated Gaussian $N_{\text{trunc}}(\cdot; \epsilon, \sigma, \epsilon + \tau, \epsilon - \tau)$ defines the probability density function on the size of perturbation vector. The smoothing function $s(\cdot; \tau, \xi, C)$ smooths the ground truth label of the perturbed datapoint, and $\xi$ determines the extent of exponential label smoothing. For the right half, smoothing function reduces the confidence in the true label in an exponential way that interpolates $(\epsilon, 1)$ and $(\epsilon + \tau, 1/C + \xi)$. To reduce the hyperparameter search space, we have used $\sigma = 0.5\tau$ and $\tau \approx \epsilon$, of which the values are determined

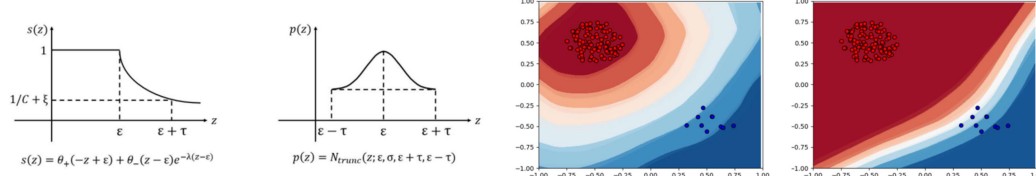

Figure 2: Illustration on the components and theoretical property of ESP. **Left:** Smoothing function $s(z)$ and probability density function $p(z)$ w.r.t perturbation size $z$. Smoothing function smooths the true label exponentially after threshold $\epsilon$. Hyperparameter $\xi$ adjusts the scale of smoothing. Probability density function defines the size of perturbation, whose maximum and minimum bound is symmetric w.r.t $x = \epsilon$. $\theta_+(z), \theta_-(z)$ are the indicator functions $\mathbb{I}[z \geq 0]$ and $\mathbb{I}[z > 0]$, respectively. **Right:** The smoothing function of ESP makes a classifier less sensitive to the number of datapoints consisting of manifolds, resulting in more reasonable decision boundary (Theorem 1 3.3.) $a = 100, b = 10, M = 10, \|\epsilon_i\| \leq 0.25$.

---

**Algorithm 1** ESP psuedocode

---

**Input:** input data $(x, y)$, noise hyperparameter $\epsilon, \sigma, \tau$, smoothness hyperparameter $\xi$
**Output:** augmented data $(x', y')$
$k \sim N_{trunc}(\epsilon, \sigma, \epsilon + \tau, \epsilon - \tau)$
$v \sim N(0, I)$
$x' \leftarrow x + \delta, \text{ where } \delta = ku, \ u = \frac{v}{\|v\|}$
$y'_i \leftarrow \begin{cases} s(k; \tau, \xi, C) & \text{if } y_i = 1 \\ \dfrac{1 - s(k; \tau, \xi, C)}{C - 1} & \text{otherwise} \end{cases}, \text{ where}$
$s(z; \tau, \xi, C) = \begin{cases} e^{-\lambda(z-\epsilon)} & \text{if } z \geq \epsilon \\ 1 & \text{otherwise} \end{cases} \text{ and } \lambda = \frac{1}{\tau - \epsilon} \ln \frac{C}{1 + \xi C}$
**return** $x', y'$

---

empirically. Two plots on the left side of Figure 2 indicate the smoothing function and the truncated Gaussian. Algorithm 1 is the pseudocode of ESP.

### 3.3 Theoretical Properties of ESP

ESP can be viewed as a generalized form of $L_2$ noise family. When $\tau \to 0$ and $\xi \to 1 - 1/C$, ESP has the same effect as $L_2$ noise with the constant perturbation radius $\epsilon$. Herein, we provide the theoretical analysis of ESP, i.e., the decision boundary of ESP is less affected by the imbalance between classes, and the relationship between the ESP's input perturbations and flat minima. When focusing on the first property, the label smoothing function of ESP makes the optimal decision boundary less affected by the number of datapoints composing manifolds, but more by the position of manifolds and the distance between manifolds. For simplicity, we provide a binary classification task with the imbalance between two clusters.

**Problem Formulation** Suppose a binary classification dataset $D$ is given as follows:

$$D = \{(\mathrm{x}, \mathrm{y})\} = \{(\mathrm{x}_1 + \epsilon_1, 1, 0), \cdots, (\mathrm{x}_1 + \epsilon_a, 1, 0), (\mathrm{x}_2 + \epsilon_{a+1}, 0, 1), \cdots, (\mathrm{x}_2 + \epsilon_{a+b}, 0, 1)\}, \quad (1)$$

where $\mathrm{x} \in \mathbb{R}^m, \mathrm{y} \in \{0, 1\}^2$. Also, $a$ and $b$ are integer and $b = aM$. Larger $M$ implies the imbalance between classes, i.e., the class $\mathrm{y} = (0, 1)$ contains $M$ times larger number of samples than the class $\mathrm{y} = (1, 0)$. $\mathrm{x}_1$ and $\mathrm{x}_2$ are the centroids of two different classes, respectively. $\epsilon_i$ indicates the deviations around the centroids. The problem setting simulates a binary classification task, where data samples are located around their class centroids and the number of samples of two classes is imbalanced by the factor of $M$. By letting $\epsilon_i \to 0$ for all $i \in [a + b]$, we can make each cluster to be concentrated, i.e., it stands for a binary classification task with the clearly separated clusters.

Then let us denote a perturbed data point without label smoothing as $(X_\delta, y_\delta) := (\mathrm{x} + \delta, \mathrm{y})$. To be specific, $(\mathrm{x}, \mathrm{y})$ is a uniformly sampled data point from $D$, and $\delta := ku, k \sim N_{\text{trunc}}(\epsilon, \sigma, \epsilon + \tau, \epsilon -$

$\tau), u := v/\|v\|, v \sim N(0, I)$, i.e., ESP without label smoothing. Similarly, let us denote a perturbed data point via ESP as $(X_E, y_E) := (\mathrm{x} + \delta, \tilde{\mathrm{y}})$, where $\tilde{\mathrm{y}} := s(\|\delta\|), s(z) := \theta_+(-z + \epsilon) + \theta_-(z - \epsilon)e^{-\lambda(z-\epsilon)}$ for some $\lambda > 0$. We will use $y_{\delta 1}$ and $y_{\delta 2}$ to denote the the first and the second elements of $y_\delta$, and similarly $y_{E1}, y_{E2}$ for $y_E$.

When $M$ becomes larger than a certain value, in other words, when the imbalance becomes severe, the perturbed sample from an input x becomes more likely to be classified into the dominant class $(0, 1)$, i.e., $y_{\delta_2} \geq 0.5$ and $y_{E_2} \geq 0.5$ for ESP without smoothing and ESP respectively. The following mathematical statement claims that ESP endures a more severe class imbalance that makes any perturbed inputs be more likely to be classified into the dominant class.

**Theorem 1.** *For any $x$, let $n' \in \mathbb{R}^+$ be the number such that for all $M \geq n'$, $\mathbb{E}[y_{\delta 2} \mid X_\delta = x] \geq 0.5$ holds, and let $n \in \mathbb{R}^+$ be the number such that for all $M \geq n$, $\mathbb{E}[y_{E2} \mid X_E = x] \geq 0.5$ holds. Then, $n \geq n'$.*

Theorem 1 claims that the smoothing function of ESP makes the optimal decision boundary, $\{x \mid \mathbb{E}[y_{E1} \mid x] = \mathbb{E}[y_{E2} \mid x] = 0.5\}$, less affected by the severity of the data imbalance, $M$. In Figure 2. we simulate a binary classification task with a strong class imbalance and show that ESP is less sensitive to the imbalance. The decision boundary of ESP locates around the middle of two manifolds, but ESP without smoothing, pushes the decision boundary to the inferior class, which makes almost all regions classified into the dominant class. Proof of Theorem 1 is on Appendix D.

In the perspective of domain generalization, researches insist that seeking flat minima in the parameter space increases model robustness against distribution shifts (Izmailov et al., 2018; Cha et al., 2021b). If perturbation in the input space can be related to the perturbation in the parameter space, we can deduce that the perturbation of ESP encourages a model to find flat minima in the parameter space. We partially formalize the property by considering a linear model with sigmoid activation function.

**Problem Formulation** Given a linear model $f : x \mapsto \sigma(Wx + b)$, where $x \in \mathbb{R}^n, W \in \mathbb{R}^{m \times n} (m < n), b \in \mathbb{R}^m$, we consider the relationship between input perturbation ($\delta \in \mathbb{R}^n$) and parameter perturbation ($\Delta \in \mathbb{R}^{m \times n}$) that satisfies $\sigma(W(x + \delta) + b) = \sigma((W + \Delta)x + b)$. When we have input perturbation bounded by $L_2$-norm, i.e., $\|\delta\| \leq \gamma$, what will be the possible perturbation region $R_\Delta$ for $\Delta$ so that for any $\|\delta\| \leq \gamma$, there exists $\Delta \in R_\Delta$ satisfying the equality or vice versa? Conversely, what will be the perturbation region $R_\delta$ for $\delta$, given $\|\Delta\| \leq \gamma$?

**Definition 1.** *(Definition of $R_\delta$) Given $W \in \mathbb{R}^{m \times n}$, $x \in \mathbb{R}^n$, and parameter perturbation region $\{\Delta \in \mathbb{R}^{m \times n} \mid \|\Delta\| \leq \gamma\}$, $R_\delta \in \mathbb{R}^n$ is a region that satisfies the following constraint:*

$$\forall \|\Delta\| \leq \gamma, \exists \delta \in R_\delta \ s.t. \ W\delta = \Delta x \text{ and } \forall \delta \in R_\delta, \exists \|\Delta\| \leq \gamma \ s.t. \ W\delta = \Delta x$$

**Definition 2.** *(Definition of $R_\Delta$) Given $W \in \mathbb{R}^{m \times n}$, $D = \{x_1, \cdots, x_N\}(x_i \in \mathbb{R}^n/\{0\} \text{ for } i \in [N])$, and input perturbation region $\{\delta \in \mathbb{R}^n \mid \|\delta\| \leq \gamma\}$, $R_\Delta \in \mathbb{R}^{m \times n}$ is a region that satisfies the following constraint:*

$$\forall x \in D, \forall \|\delta\| \leq \gamma, \exists \Delta \in R_\Delta \ s.t. \ W\delta = \Delta x \text{ and } \forall x \in D, \forall \Delta \in R_\Delta, \exists \|\delta\| \leq \gamma \ s.t. \ W\delta = \Delta x$$

With these definitions on the regions of interest, we now present theorems on converting perturbations in input space to parameter space (Theorem 2) and vice versa (Theorem 3, 4.) Colloquially, Theorem 2 states that perturbations bounded by $L_2$ norm in the parameter space can be converted to the perturbations bounded by an rotated ellipsoid in the input space. Meanwhile, converting the perturbation in the input space to parameter space in a closed form expression is infeasible. As an alternative, we provide the subset and the superset of the converted perturbation region in the hyperparameter space in Theorem 3 and 4. $X_\lambda$ is an $(m \times n)^2$ square matrix which is defined by the input $x$ and weight $W$. The formal definition on $X_\lambda$ is stated in Appendix C.

**Theorem 2.** *Given $W \in \mathbb{R}^{m \times n}$, $x \in \mathbb{R}^n$, and parameter perturbation region $\{\Delta \in \mathbb{R}^{m \times n} \mid \|\Delta\| \leq \gamma\}$, a volume-zero m-dim rotated ellipsoid satisfies the definition of $R_\delta$.*

**Theorem 3.** *Given $W \in \mathbb{R}^{m \times n}$, $D = \{x_1, \cdots, x_N\}(x_i \in \mathbb{R}^n/\{0\} \text{ for } i \in [N])$, and input perturbation region $\{\delta \in \mathbb{R}^n \mid \|\delta\| \leq \gamma\}$, let $x_{max} := \arg\max_{x_i} \|x_i\|$ and $\lambda_{min} := \min\{\lambda_1, \cdots, \lambda_m\}$. $\{\Delta \in \mathbb{R}^{m \times n} \mid \|\Delta\| \leq (\|x_{max}\|^2/\lambda_{min}^2)^{-1}\} \subseteq R_\Delta$*

**Theorem 4.** *Given $W \in \mathbb{R}^{m \times n}$, $D = \{x_1, \cdots, x_N\}(x_i \in \mathbb{R}^n/\{0\} \text{ for } i \in [N])$, and input perturbation region $\{\delta \in \mathbb{R}^n \mid \|\delta\| \leq \gamma\}$, let $R_i := \{d \in \mathbb{R}^{m \times n} \mid d^\top X_\lambda^{(i)} d \leq 1\}$ and $\Gamma := \{R_i \mid i \in [N]\}$. $R_\Delta \subseteq \{\arg\min_{R_1, \cdots, R_n \in \Gamma} \max_{\rho \in \cup_{i \in [n]} R_i} \|\rho\|^2\}$.*

ESP introduces a bounded noise into input samples, and it means that the perturbed samples can deviate in the bounding sphere around the clean input sample. Also, ESP smoothly decays the confidence level of labels for the perturbed inputs in the bounding sphere. The aforementioned mathematical statements claim that the perturbed inputs in a bounded sphere can be interpreted as the perturbed parameter space in a bounded ellipsoid for the clean input samples, and vice versa. Consequently, label smoothing of ESP makes the perturbations of parameters in the bounded ellipsoid smooth labels of the clean input samples. Then the loss landscape in the parameter space becomes more smooth by preventing the rapid change of loss values due to the perturbations of parameters. Along with the theoretical analysis, we empirically confirm that ESP achieves flatter minima than the baselines. The proofs of Theorem 2, 3, and 4 are on Appendix B and C.

## 4 EXPERIMENTS

Table 1: Model robustness over MNIST-C, CIFAR-10/100-C, and Tiny-ImageNet-C benchmarks in the measure of average corruption error (lower is better). The reported values are the average corruption error of three individual runs for each method. The best results are marked in bold.

| Augmentation | MNIST-C | CIFAR-10-C | CIFAR-100-C | Tiny-IN-C |
|---|---|---|---|---|
| Naive | $8.01 \pm 0.10$ | $25.57 \pm 0.45$ | $52.21 \pm 0.47$ | $75.49 \pm 0.24$ |
| Naive + $L_2$ | $7.07 \pm 0.43$ | $18.55 \pm 0.26$ | $45.64 \pm 0.11$ | $75.09 \pm 0.15$ |
| **Naive + ESP** | $\textbf{6.45} \pm 0.02$ | $16.17 \pm 0.41$ | $40.28 \pm 0.29$ | $73.97 \pm 0.37$ |
| AugMix | $14.36 \pm 0.30$ | $10.67 \pm 0.09$ | $35.50 \pm 0.10$ | $67.78 \pm 0.48$ |
| AugMix + $L_2$ | $12.02 \pm 0.29$ | $10.36 \pm 0.07$ | $35.12 \pm 0.20$ | $67.81 \pm 0.01$ |
| **AugMix + ESP** | $11.69 \pm 0.29$ | $\textbf{8.62} \pm 0.11$ | $34.59 \pm 0.18$ | $67.71 \pm 0.03$ |
| DeepAugment | $10.68 \pm 0.27$ | $13.21 \pm 0.11$ | $39.54 \pm 0.04$ | $64.75 \pm 0.32$ |
| DeepAugment + $L_2$ | $10.57 \pm 0.06$ | $11.94 \pm 0.21$ | $38.82 \pm 0.29$ | $64.58 \pm 0.33$ |
| **DeepAugment + ESP** | $10.35 \pm 0.52$ | $11.16 \pm 0.07$ | $36.46 \pm 0.21$ | $61.43 \pm 0.07$ |
| AugMix + DeepAug | $7.45 \pm 0.65$ | $9.15 \pm 0.06$ | $32.56 \pm 0.05$ | $60.61 \pm 0.13$ |
| AugMix + DeepAug + $L_2$ | $7.22 \pm 0.05$ | $9.01 \pm 0.09$ | $32.44 \pm 0.06$ | $61.07 \pm 0.19$ |
| **AugMix + DeepAug + ESP** | $7.09 \pm 0.33$ | $8.90 \pm 0.11$ | $\textbf{32.23} \pm 0.17$ | $\textbf{59.02} \pm 0.21$ |

### 4.1 DATASET STATISTICS

**MNIST-C (Mu & Gilmer, 2019)** 15 corruptions (brightness, canny edges, dotted line, fog, glass blur, impulse noise, motion blur, rotate, scale, shear, shot noise, spatter, stripe, translate, zigzag). There are 10,000 images corresponding to each corruption, resulting in total 150,000 images.

**CIFAR-10/100-C, Tiny-ImageNet-C (Hendrycks & Dietterich, 2019)** 15 corruptions (brightness, contrast, defocus blur, elastic transform, fog, frost, Gaussian, glass, impulse noise, jpeg compression, motion blur, pixelate, shot noise, snow, zoom blur), 5 severities. There are 10,000 images corresponding to each severity, resulting in total 750,000 images.

### 4.2 EXPERIMENTAL SETUP

**Model Architecture** For MNIST-C benchmark, we have used convolutional neural network architecture proposed in (Rony et al., 2019). For CIFAR-10/100-C benchmarks, we have used WRN-40-2 model (Zagoruyko & Komodakis, 2016) as backbone network. For Tiny-ImageNet-C benchmark, ResNet18 (He et al., 2016) has been employed.

**Optimizer** In all our experiments, SGD momentum with initial learning rate of 0.1 and momentum value of 0.9 has been used. For both MNIST-C and CIFAR-10/100-C experiments, we have used cosine learning rate decay scheduling to train the model until convergence as in Hendrycks et al. (2021b). For Tiny-ImageNet-C benchmark, we have utilized step learning rate decay scheduling at 100 and 150 epoch with the coefficient of 0.1 as in Wang et al. (2021b).

**Hyperparameter Tuning** We have used grid search to find the optimal hyperparameters for $L_2$ noise and ESP. As mentioned in Section 3.2, we use $\sigma = 0.5\tau$ and $\tau \approx \epsilon$ to reduce the hyperparameter search space. Despite the fact that there is no general rule for deciding $\xi$ values, we have chosen $\xi$ such that the maximally smoothed true label $(C^{-1} + \xi)$ is $\gamma$ times higher than the other labels $((1 - C^{-1} - \xi)/(C - 1))$. Specifically, we have chosen $\gamma = \{10, 20\}$ for MNIST-C/CIFAR-10-C, $\gamma = \{20, 50, 100\}$ for CIFAR-100-C, and $\gamma = \{200\}$ for Tiny-ImageNet-C. Such choice of gamma results in $\xi = \{0.426, 0.590\}$ for MNIST-C/CIFAR-10-C, $\xi = \{0.158, 0.326, 0.493\}$ for CIFAR-100-C, and $\xi = \{0.496\}$ for Tiny-ImageNet-C.

**Evaluations** As in Hendrycks et al. (2021b), we have calculated the average corruption error across different corruption types and severities.

Further experiment details can be found at Appendix E.

### 4.3 RESULTS

We first examined the performance of ESP with respect to different data augmentations in common corruption benchmarks (referring Table 5.) On the MNIST-C benchmark, AugMix and DeepAugment impaired model robustness in contrast to $L_2$ noise and ESP. The ensemble of AugMix and DeepAugment slightly increased model robustness, but was still inferior to $L_2$ noise and ESP. On the other hand, $L_2$ noise and ESP increased model robustness either as a standalone method or combined with other methods. In the CIFAR-10/100-C experiment, all methods improved model robustness both solely and in composition with other methods as well. On the CIFAR-10-C benchmark, AugMix enhanced model robustness the most as a sole method, and AugMix composed with ESP enhanced robustness the most as an ensemble method. In CIFAR-100-C experiment, AugMix exhibited the best performance among the sole methods, and combining AugMix, DeepAugment, and ESP together yielded the highest robustness among ensemble methods. In Tiny-ImageNet-C experiment, ESP boosted model robustness in all circumstances by large margin, in constrast to $L_2$ noise which had trivial or no improvement on the model robustness. In general, ESP showed consistent improvement on the model's robustness in all experiment cases, dominating $L_2$ distance based noise.

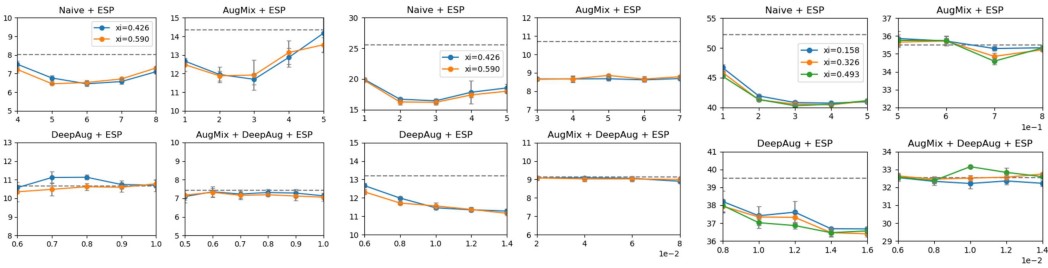

Figure 3: Model robustness across different hyperparameter configurations consisting of ESP's search space in MNIST-C (left), CIFAR-10-C (middle), and CIFAR-100-C (right) dataset. X-axis represents the $\epsilon$ value of ESP, and y-axis represents the error according to varying $\epsilon$. The average corruption error of original augmentation method is represented as a gray line.

Next, we compared the model performance over different hyperparameters consisting the search space of ESP (Figure 3.) While ESP improved model robustness in most cases, the amount of robustness gain differed meaningfully with respect to the perturbation size ($\epsilon$) and its corresponding hyperparameters ($\tau, \sigma$) in many cases, drawing a convex average corruption error loss graph with respect to the perturbation size. One interpretation of the convex-shaped loss graph is that when the perturbation is too small, model learns a decision boundary that is not general enough; however when the perturbation is too large, the decision boundary tumbles down due to fuzzy data augmentations intruding each other's manifolds severely.

Subsequently, we investigated the flatness of each augmentation method in the CIFAR-10-C benchmark (Figure 4). With varying radius, we used Monte-Carlo simulation with 50 individual samples for each method. The input perturbation that has bounded $L_2$ norm of ESP encouraged the model to find a relatively flat minimum in the hyperparameter space compared to the naive Empirical Risk

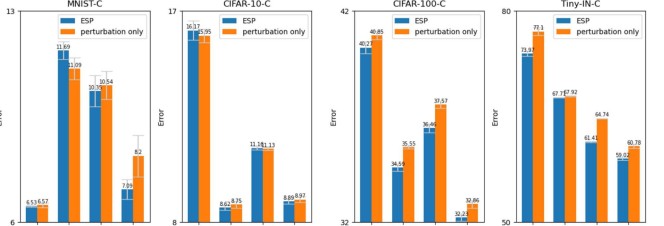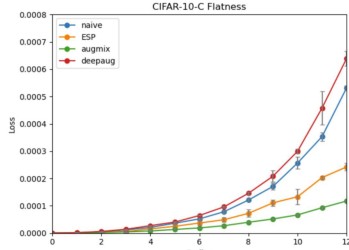

Figure 4: Experiment results on the removal of smoothing function and on the flatness of local minima. **Left:** An ablation study on the smoothing function of ESP. **A+D** denotes the ensemble of AugMix and DeepAugment methods. **Right:** The flatness of Empirical Risk Minimization (denoted as naive), ESP, AugMix, and DeepAugment, respectively in CIFAR-10-C benchmark.

Minimization (ERM). While AugMix found the flattest minimum in the parameter space, the local minimum found by DeepAugment was escalating the most.

Finally, we analyzed how the removal of the smoothing function in ESP can affect performance. In MNIST-C and CIFAR-10-C benchmarks, there were no statistically meaningful differences in the model robustness. Nevertheless, removing the smoothing function of ESP significantly harmed model robustness in CIFAR-100-C and Tiny-ImageNet-C benchmarks. One possible explanation for this phenomenon is that the increase in the diversity of classes results in smaller and diverse data manifolds with different labels. This may induce larger overlapping areas of perturbations with different labels in the input space.

## 5 DISCUSSION

AugMix and DeepAugment damaged robustness in MNIST-C benchmark, but prominently enhanced model robustness in CIFAR-10/100-C and Tiny-ImageNet-C benchmark. On the other hand, ESP showed the tendency to consistently improve model robustness in a mild way. We interpret this phenomenon as the difference between each method's inductive bias. Since ESP is a high-level data-agnostic algorithm, the robustness gain of data augmentation may not be drastic compared to the existing methods. Nonetheless, there is more room for exploiting ESP, regardless of the semantics of dataset.

However, ESP is sensitive to the choice of hyperparameter that determines the maximal perturbation size. The problem stems from the intrinsic nature of perturbation based augmentation methods. With varying data distributions on different tasks, we cannot estimate the sweet spot of ESP before actually conducting model training with varying perturbation sizes. Insufficient perturbations will trivially improve robustness, while intense perturbations will demolish the decision boundary of the target model due to manifold intrusions overwhelming in the end.

## 6 CONCLUSION

Deep neural networks (DNNs) being prone to real-world driven common data corruptions, various methods have been proposed to increase model robustness. Among several approaches, we have focused on developing augmentation-based method due to its broad applicability. Inspired by the robustness gain achieved by simple $L_2$ distance based random noise, we have proposed an efficient and general data augmentation method, ESP, that makes classifier robust to diverse image data corruptions without strong inductive bias on the nature of dataset. The data augmentation nature of ESP enforces a classifier to have a contour-like decision boundary, different from most of the existing DNN learning algorithms. Moreover, we have provided theoretical analysis and experiment result on how perturbations with bounded $L_2$ norm can be related to the perturbations in the parameter space. Despite the fact that we have only exploited corrupted image classification benchmarks on measuring the robustness gain, ESP can be exploited to different classification tasks other than image classification to enhance a model's robustness to unexpected data corruptions.

