# OpenReview forum: "ESP: Exponential Smoothing on Perturbations for Increasing Robustness to Data Corruptions"
_ICLR.cc/2023/Conference — Submitted to ICLR 2023_

### Official Review · Reviewer_ST3E · 2022-10-24

**Confidence:** 4
**Correctness:** 3
**Technical Novelty And Significance:** 3
**Empirical Novelty And Significance:** 3
**Recommendation:** 5

**Clarity, Quality, Novelty And Reproducibility:**

The overall presentation remains to be improved. The novelty of the proposed method is limited. The theoretical justification might be a solid contribution. Its significance cannot be told since the appendix is missing.

**Strength And Weaknesses:**

Strength:
1. The proposed data augmentation is simple, effective, and also theoretically justified.
2. Both sample-specific adversarial perturbation and universal adversarial perturbation are considered in this study.

Weakness:
1. As also pointed out by the author, the proposed method is sensitive to hyperparameters. The reported improvement in Table is often marginal. The two facts make the proposed method less practical.
2. Given the nature of the proposed data augmentation, the author should report the mean and variance of each experiment since randomness is brought by sampling from a truncated Gaussian distribution. This is necessary since the improvements reported are not significant.
3. In Figure 4, the authors aim to claim the effectiveness of exponential label smoothing. However, the standard label smoothing should be included as a baseline. The improvement of ESP over perturbation-only baseline can be brought by the effectiveness of standard label smoothing.
4. In Figure 2, the theoretical properties of ESP are illustrated. Is it possible to draw the illustration with real experimental data?
5. There are many appendix points in the main text. However, no appendix is found in the submission.


**Summary Of The Paper:**

The authors propose a data-agnostic and model-agnostic data augmentation method ESP to enhance a model’s robustness to unforeseen data corruptions. Concretely, ESP consists of two parts, perturbation adding and exponential label smoothing. The added noise perturbation in ESP is sampled from a truncated Gaussian distribution, while Label smoothing here takes the magnitude of perturbation into consideration. Stronger perturbations are companioned by stronger label smoothing.
The author provides theoretical proof that perturbations in input space can encourage a model to find a flat minimum on the parameter space.
Experiments demonstrate that ESP when combined with existing augmentation methods improves the robustness to different degrees.


**Summary Of The Review:**

Given the weakness listed above, I tend to reject this paper before rebuttal.

---

> ### Author Response · Authors · 2022-11-18
>
> 1. Marginal performance gain over the cost of hyperparameter tuning.
> - We want to emphasize that ESP shows consistent robustness gains over all cases when combined with other prior methods. In contrast, AugMix and DeepAugment show significant degradation of model robustness by -6.35% and -2.67% on the MNIST-C digit classification benchmark when compared with the ‘Naïve’ approach, while ESP improved robustness by +1.66%. We conjecture that the augmentations of AugMix and DeepAugment with strong inductive bias are sensitive to the semantics of the dataset, even more severe in a small dataset such as MNIST-C. However, ESP is a data-agnostic method that has not been explored by the cutting-edge method, so it can bring consistent and robust gains over all combination cases. Based on these reasons, we want to argue that the performance gain of ESP is not marginal.
>
> 2. The average and standard deviation seems missing.
> - We have made three individual runs per each method, and recorded the average and standard deviations in the main table (Table 1). In addition, we have drawn error bars at every plotting in our paper (Figure 3, 4.) If you find any statistics with missing mean and standard deviation, please feel free to point it out and let us know.
>
> 3. Standard label smoothing vs ESP
> - Standard label smoothing won't increase model robustness since the label smoothing function we have introduced is a tool to dynamically embed the perturbations' uncertainties in the input space. We partially share the experiment results of standard label smoothing in our official comment.
>
> 4. Can the plotting of ESP be made on experimental data?
> - There were two reasons we have exploited spiral and spherical 2D toy datasets: i) For these toy examples, we do not have to worry about the distortion in dimensionality reduction when trying to visualize data samples in 2D space, and ii) it is infeasible to plot the decision boundary in high dimension. To be more precise with the second reason, the decision boundaries are plotted by making a 2D-grid and plotting the model's predictions at each coordinate. When it comes to image data, e.g. MNIST with 28 x 28 dimensions, the computation cost is $\Omega (2^{28*28})$. While we can map the datapoints themselves into 2D using tSNE, UMAP, etc., depicting the decision boundary is infeasible.
>
> 5. Where is Appendix?
> - The Appendix is located at the "Supplementary Material" section. Please download the .zip file and unzip the supplementary pdf file. (We have checked that the file successfully opens in both Windows and MacOS environment.)

---

### Official Review · Reviewer_6X1k · 2022-10-24

**Confidence:** 4
**Correctness:** 3
**Technical Novelty And Significance:** 3
**Empirical Novelty And Significance:** 3
**Recommendation:** 6

**Clarity, Quality, Novelty And Reproducibility:**

This paper is written in good quality but needs some improvement in the theoretical properties part.

**Strength And Weaknesses:**

Strength:
* The theoretical analysis provides some insights on the benefits of label smoothing.
* The proposed ESP augmentation has the potential to be combined with many existing works on common corruption robustness to further boost the model robustness.

Weaknesses:
* The theoretical analysis is not clear enough. The imbalance's severity N and the normal distribution use the same letter N, which makes me confused when reading this section.
* Theorem 1 states that ESP can make the optimal boundary less sensitive to imbalance severity, but why is this a desired property for ESP? What is the benefit for a less-sensitive boundary?
* Theorem 2 proves that perturbations in the input space can be converted to perturbations in parameter space. This result is available in [1] Lemma A.4.
* In experiments, the paper only compares the result with vanilla L2. I suggest the authors to compare with more label smoothing methods and also mixup.

[1] Convergence of Adversarial Training in Overparametrized Neural Networks


**Summary Of The Paper:**

This paper proposed a new augmentation strategy ESP, which augment training with random perturbation with smoothed labels. The experiments showed that ESP outperforms vanilla L2 augmentation on common corruption robustness. The authors also gave theoretical analysis on the properties of ESP.

**Summary Of The Review:**

In all, I think this paper proposed a simple yet effective method to boost the model robustness against common corruption. The theoretical analysis also provides some insights. However, as I have mentioned in the Weaknesses, I hope the authors can have more experimental comparison and also compare with some related existing theoretical results.

---

> ### Author Response · Authors · 2022-11-18
>
> 1. The theoretical analysis are not clear enough. (especially the symbol for normal pdf N and imbalance severity N)
> - We will replace the syntax for the imbalance severity, which may have confused readers. In addition, we have posted additional intuitions of our theorems. We hope that it clarifies the meaning of the theoretical analysis. Please check out to see if the modified version of our paper and the below explanations resolves your concerns. Otherwise, please leave us comments We will be happy to reply back.
>
> 2. Why is Theorem 1 a desirable property of ESP?
> - First, the data imbalance problem is quite common in real-world environments. Theorem 1 states that ESP’s smoothing function makes the optimal decision boundary be less shaky or prone to the data imbalance severity. Perturbation-based methods without our smoothing function are shown to be vulnerable to the data imbalance problem (please see Figure 2 Right `w/o smoothing case’), while ESP with smoothing suppresses the problem by forming the decision boundary at the middle ground between the manifolds from different classes. Second, even with a balanced experimental dataset, the diverse topology of manifolds can be formed in the input space, where each manifold contains a different number of samples. The smoothing function of ESP dampens the effect of both the manifold intrusion problem and data imbalance problem at the manifold level, making the optimal decision boundary be less biased.
>
> 3. There is already a proof on mapping input perturbations to parameter space.
> - We truly appreciate your notification of the prior work that shows a similar idea as we did. Leaving minor differences aside, the major difference between Lemma A.4 of the work and our Theorem 3 & 4 (input to parameter conversion) is that the parameter region in Lemma A.4 is not bounded (in our work, bounded by $R_\Delta$.) Simply speaking, we have shown that L2-norm bounded input perturbation can be mapped to another bounded parameter perturbation by providing the bounded superset and subset of parameter perturbation region $R_\Delta$ ($R_{sup}$ and $R_{sub}$ in our work, respectively.) On the other hand, Lemma A.4. only induces that parameter perturbation region includes some L2-norm bounded subset ($R_{sub}$.) – which is not sufficient to conclude that all the bounded input perturbations are mapped to the bounded parameter space. Formally speaking, Lemma A.4 proves that $\exists \| \Delta \|_F \leq c $ such that $W\delta = \Delta x$. With this sole constraint, $R_\Delta$ is not bounded since parameter space is larger than input/output space; thus, we have introduced additional constraints over the dataset $D=\{x_1, \cdots, x_N\}$. Given $\|\delta\| \leq \gamma$, for any $c \propto \gamma$ we can always find $\Delta$ s.t. $W\delta = \Delta x$ and $\|\Delta \|_F > c$.
>
> 4. Comparison of ESP with MixUp and other label smoothing method?
> - The experiment results from AugMix show that MixUp is inferior to both AugMix and our method (Table 1, WideResNet). Furthermore, the authors of AugMix stated that combining AugMix with MixUp deteriorated model robustness, possibly due to the manifold intrusion problem. (For more details, please refer to the comment we've made to the first reviewer since the question seems to be fairly overlapping.) Standard label smoothing does not increase model robustness since the label smoothing function we have introduced is a tool to dynamically embed the perturbations' uncertainties in the input space. We partially share the experiment results of standard label smoothing in our official comment.

---

> > ### Comment · Reviewer_6X1k · 2022-11-23
> > **Thank you for your response**
> >
> > Thank you for your detailed reply. Most of my questions have been addressed but I still hold some concerns on the contribution of this paper. The idea of reweighing perturbed samples is used in many adversarial robustness papers (though different from robustness against common corruption). Also, part of the theorem has been introduced before.
> > Therefore, I keep my recommendation that this paper is on the borderline and I raised the score to 6.

---

### Official Review · Reviewer_rCik · 2022-10-25

**Confidence:** 3
**Correctness:** 3
**Technical Novelty And Significance:** 3
**Empirical Novelty And Significance:** 2
**Recommendation:** 5

**Clarity, Quality, Novelty And Reproducibility:**

The paper is overall mostly clearly written, and provides an original method for label smoothing based on perturbation size.

**Strength And Weaknesses:**

Strengths:
- The paper is mostly well written, and the contributions/methods are clear.
- The paper provides some theoretical justifications for their method.
- The proposed method results in consistent improvements in generalization on common corruptions across different datasets.

Weaknesses:
- Since the proposed method is augmenting the data with noise, and evaluating on the common corruptions dataset, of which several of the corruptions are noise-based corruptions, I think it's important to state whether robustness to all of the corruption types improves, or just those noise type corruptions.
- The authors mention the proposed method is sensitive to the choice of hyperparameter determining the max perturbation size. When doing the hyperparameter search, how do you determine the optimal hyperparameters? I.e. are you choosing the hyperparameters based on performance on the common corruptions test set? In practice, you'd need some other validation metric for choosing hyperparameters, as fundamentally you are trying to improve performance on unseen, out of distribution data.
- The paper does not report performance on clean/unperturbed test images. How does the proposed data augmentation scheme affect standard test accuracy?

Other suggestions:
- I would maybe state earlier in the paper that the label smoothing technique only results in improvements for datasets with a larger number of classes.
- Most of the figures are too small to properly read. It's a bit difficult to gather any insights from Figure 3.


**Summary Of The Paper:**

This paper proposes a method for improving robustness to common corruptions via L2-norm bounded data augmentations of varying sizes. The authors show theoretically that data augmentation of this form results in models with flatter minima in parameter space, which has been shown to result in better generalization. This paper also proposes a method for label smoothing that exponentially smooths the labels based on the size of the perturbation.

**Summary Of The Review:**

While this paper does report improvements across all datasets in terms of the common corruptions performance and includes some theoretical justifications, it's unclear to me whether the proposed method results in performance improvements for a variety of different unseen perturbations, or just noise-based perturbations. It is also unclear how the hyperparameters would be chosen in practice for unseen perturbations. Given these uncertainties, I'm leaning towards reject.

---

> ### Author Response · Authors · 2022-11-13
>
> 1. Since ESP is perturbation(noise)-based method, does it generally increase robustness over all the corruption types or limited to noise-based corruptions?
> - The benchmarks considered here already evaluate the robustness to various kinds of corruption types, e.g., CIFAR-100-C considers (add the types of corruptions) as the unseen corruptions. Therefore, the gains of ESP for the benchmarks mean that ESP enhances the robustness to various kinds of corruption types, not limited to noise-based corruption. Because the test accuracies in the main paper are the average over the various types of corruption, we hope to add individual accuracy for each type of corruption until the revision deadline.
>
> 2. How is hyperparameter search done?
> - Let us clarify our hyperparameter selection. For each hyperparameter choice, we train a model for a certain hyperparameter choice until the model converges by using cosine learning rate decay (we only use source (clean) training data). Then among the choices, we report the case with the best accuracy in the target (corrupted) data. We point out that there is no validation set in the corruption benchmarks. For this case, it is fair to choose the hyperparameters with the best test accuracies on clean data. However, we made a minor mistake in choosing the hyperparameters with the best test accuracy on corruption data. Albeit the mistake, we want to say that in many cases, the model with the lowest error on the clean test set also had the lowest error on the corrupted dataset. During the rebuttal period, we are willing to report the corrected results in our final paper.
>
> 3. No report on clean accuracy.
> - We will add the clean accuracy until the end of the revision deadline ASAP. In general, there were no strong tendency to either improve or impair; augmentations resulted in roughly the same clean accuracy.

---

> > ### Author Response · Authors · 2022-11-19
> > **Experimental results on the first concern**
> >
> > Since ESP is perturbation(noise)-based method, does it generally increase robustness over all the corruption types or limited to noise-based corruptions?
> > - We have reproduced the results on MNIST-C and CIFAR-10/100-C to record and find out detailed statistics. Among 15 different corruptions, *impulse noise* and *shot noise* have been considered as noise-based corruptions on MNIST-C benchmark and *gaussian, shot noise, impulse noise* on CIFAR-10/100-C benchmark.
> > With three benchmarks and four baseline methods (Naive, AugMix, DeepAugment, AugMix + DeepAugment), there are total 12 experimental cases examining the effect of ESP on model robustness in the aspect of noise based and non-noise based corruptions. We have run three individual trials per each case and averaged the noise based corruptions and non-noise based corruptions. The robustness against noise-based corruptions has been increased by ESP on 11 out of 12 cases, and the other corruptions 9 out of 12 cases. Overall, there were absolute increase in the average corruption error in all the cases. This indicates that ESP makes a classifier robust against both noise-based and other types of image corruptions, with stronger robustness on noise-based corruptions.

---

### Official Review · Reviewer_eTKu · 2022-10-29

**Confidence:** 4
**Clarity, Quality, Novelty And Reproducibility:** Please see above
**Correctness:** 2
**Technical Novelty And Significance:** 2
**Empirical Novelty And Significance:** 2
**Recommendation:** 3

**Strength And Weaknesses:**

1. The isotropic random noise might not be the best choice for image augmentation. Imagine two random noises with the same magnitude, one is focused on the foreground object, while the other is focused on the background pixels. The former on should be accompanied with a larger perturbation on the label (i.e. a more smoothed label in your framework), while the later should be accompanied with a smaller label perturbation. This is because the background noise hardly affects the semantic meaning of the image, and thus the label shouldn't change much. However, in your current design, both noises would have a same level of label perturbation, which might not be the best design in my opinion.

2. Besides the marginal performance gain, the proposed method also suffers from hyper-parameter tuning overhead. Specifically, it introduces four new hyper-parameters to tune. I wonder whether it is worthy to pursue the marginal performance at such cost.

3. Since the performance gain on the small datasets are maringal, I wonder whether it brings benefits on larger datasets such as ImageNet(-C).

4. Has the performance on clean test sets been reported in the main text? Both clean accuracy and robustness are important for a real-world model. Previous works such as AugMix and DeepAug usually report clean accuracy alongside robustness.

5. It seems to me that MixUp/CutMix/etc. also share similar idea to jointly perturb label and input. Maybe the author should discuss why the proposed way is better than those previous methods and provide empirical comparison results?

Overall, I suggest revision and resubmit considering the above limitations.

**Summary Of The Paper:**

This paper proposes a new data augmentation method, which jointly augments data and label, using an empirical hand-crafted rule: The input image is added a random Gaussian noise, and the corresponding label is smoothed with the strength proportional to the image noise magnitude. I think the goal of such data augmentation is to enforce model to have a certain level of smoothness, as regularized by the strength of the augmentation. And ideally improve model robustness. However, the empirical results shows marginal improvements. For example, on CIFAR0-C and CIFAR100-C, it only brings 0.39% and 0.21% accuracy gain over the second best method. I have some assumptions on why the proposed method not bringing good performance gains. Please see details below.

**Summary Of The Review:**

Please see above

---

> ### Author Response · Authors · 2022-11-13
> **Re: Official Review of Paper2534 by Reviewer eTKu**
>
> 1. The isotropic random noise might not be the best choice for image augmentation. There may be room for image specific perturbation methods.
> - First, we truly appreciate your suggestion for our work. It seems to be an effective way to put much more robustness in capturing the semantic information of images. However, we want to note that our ESP is designed for a data- and model-agnostic method to improve the deep models’ robustness. In contrast, AugMix is based on the operations exploited in AutoAugment of [AutoAug’19], which trains reinforcement learning to find the best augmentations for a given dataset. Also, DeepAugment utilizes a parameterized module for adopting perturbations. The methodological versatility of ESP comes from random noise, which is unbiased perturbation that can work as noise signals for any form of data. We believe that the advantage makes ESP show consistent performance gains over the diverse datasets. However, AugMix and DeepAugment perform worse than the ‘Naïve’ case in the MNIST-C benchmark. We believe that your suggestion, which lays more inductive bias on a given dataset, would be the next branch of our work which is not trivial: A conjunction of ESP with a separator for the semantic (foreground) and spurious (background) parts without pixel-level labeling does seem to be an incremental job of this work. We hope to consider the issue in the near future.
> [AutoAug’19] E. D. Cubuk et al., “AutoAugment: Learning Augmentation Strategies from Data,” CVPR 2019.
>
> 2. Marginal performance gain over the cost of hyperparameter tuning.
> - We want to emphasize that ESP shows consistent robustness gains over all cases when combined with other prior methods. In contrast, AugMix and DeepAugment show significant degradation of model robustness by -6.35% and -2.67% on the MNIST-C digit classification benchmark when compared with the ‘Naïve’ approach, while ESP improved robustness by +1.66%. We conjecture that the augmentations of AugMix and DeepAugment with strong inductive bias are sensitive to the semantics of the dataset, even more severe in a small dataset such as MNIST-C. However, ESP is a data-agnostic method that has not been explored by the cutting-edge method, so it can bring consistent and robust gains over all combination cases. Based on these reasons, we want to argue that the performance gain of ESP is not marginal.
>
> 3. ESP on ImageNet-C?
> - Although we could not perform all the extensive experiments on ImageNet-C as we did in MNIST, CIFAR, and Tiny-ImageNet due to our limited resources, we have shown a minimal experiment result on ImageNet-C to verify that ESP works well in ImageNet-C as well (please see Section F of Appendix). ESP shows +9.96% of gain over Naïve. We want to point out that ESP shows a much larger performance gain when compared with the +1.52% gain on Tiny-ImageNet-C, which is a sub-dataset of ImageNet. It partially shows that ESP is more effective when the ImageNet-based benchmark becomes more complicated.
>
> 4. No report on clean accuracy.
> - We will add the clean accuracy until the end of the revision deadline ASAP. In general, there were no strong tendency to either improve or impair; augmentations resulted in roughly the same clean accuracy.
>
> 5. Comparison with MixUp and CutMix?
> - When we borrow the observations from the work of AugMix [AugMix’20], it turns out that CutMix did not improve model robustness at all, and Mixup is inferior to AugMix (Please see Table 1 in [AugMix’20]). When compared with ESP, CutMix decreased model robustness by -0.2%, and Mixup increased by +4.5%, whereas our method ESP increased by +9.40% in the CIFAR-10-C benchmark. In the CIFAR-100-C benchmark, CutMix and Mixup increased robustness by +0.4% and +2.9%, while ESP increased model robustness by +11.93%. Furthermore, the authors of AugMix stated that combining AugMix with MixUp deteriorated model robustness, possibly due to the manifold intrusion problem. Borrowing the authors’ quote, “... applying AUGMIX on top of Mixup increases the error rate to 13.3%, possibly due to an increased chance of manifold intrusion (Guo et al., 2019).“. In contrast, combining AugMix with ESP resulted in an error rate of 8.62%. ESP is relatively free from intrusion problems since the perturbation is based on a datapoint (not an interpolation between points) and has a dynamic label smoothing function that dampens the effect of manifold intrusions.
> [AugMix’20] D. Hendrycks et al., “AUGMIX: A SIMPLE DATA PROCESSING METHOD TO IMPROVE ROBUSTNESS AND UNCERTAINTY,” ICLR 2020. https://arxiv.org/pdf/1912.02781.pdf

---

### Author Response · Authors · 2022-11-19

[Standard Label Smoothing] [1]

Average Corruption Error

<MNIST-C>

Naive 10.89 $\pm$ 0.07 AugMix 14.15 $\pm$ 0.39 DeepAug 10.19 $\pm$ 0.38

<CIFAR10-C>

Naive 26.09 $\pm$ 0.48 AugMix 10.67 $\pm$ 0.16 DeepAug 13.59 $\pm$ 0.01

[Clean Accuracy]

We have added the clean error results at the end of Appendix E.


When the standard label smoothing, which is a static label smoothing, is used together with the baseline methods (Naive, AugMix, DeepAugment), it slightly harms the robustness in most cases. ESP seems to dominate the static label smoothing method in improving model robustness.


[1] Szegedy, et al. *Rethinking the Inception Architecture for Computer Vision*. CVPR 2016

---

### Decision · Program_Chairs · 2023-01-20

**Decision:**

Reject

**Justification For Why Not Higher Score:**

Weakness:

"the proposed method is sensitive to the choice of hyperparameter.";
"Specifically, it introduces four new hyper-parameters to tune. I wonder whether it is worthy to pursue the marginal performance at such cost."

There were also issues on the limited improvement for clean accuracy.

**Justification For Why Not Lower Score:**

N/A

**Metareview: Summary, Strengths And Weaknesses:**

Strength:
"--- The paper is mostly well written, and the contributions/methods are clear.
--- The paper provides some theoretical justifications for their method.
--- The proposed method results in consistent improvements in generalization on common corruptions across different datasets."

Weakness:
"the proposed method is sensitive to the choice of hyperparameter."; "Specifically, it introduces four new hyper-parameters to tune. I wonder whether it is worthy to pursue the marginal performance at such cost."

There were also questions on the improvement for clean accuracy.